# The LoCA Regret: A Consistent Metric to Evaluate Model-Based Behavior in Reinforcement Learning

**Harm van Seijen**[1], **Hadi Nekoei**[2], **Evan Racah**[2], **Sarath Chandar**[2,3,4]

[1]Microsoft Research Montréal, [2]Mila - Quebec AI Institute,
[3]École Polytechnique de Montréal, [4]Canada CIFAR AI Chair

## Abstract

Deep model-based Reinforcement Learning (RL) has the potential to substantially improve the sample-efficiency of deep RL. While various challenges have long held it back, a number of papers have recently come out reporting success with deep model-based methods. This is a great development, but the lack of a consistent metric to evaluate such methods makes it difficult to compare various approaches. For example, the common single-task sample-efficiency metric conflates improvements due to model-based learning with various other aspects, such as representation learning, making it difficult to assess true progress on model-based RL. To address this, we introduce an experimental setup to evaluate model-based behavior of RL methods, inspired by work from neuroscience on detecting model-based behavior in humans and animals. Our metric based on this setup, the *Local Change Adaptation (LoCA) regret*, measures how quickly an RL method adapts to a local change in the environment. Our metric can identify model-based behavior, even if the method uses a poor representation and provides insight in how close a method's behavior is from optimal model-based behavior. We use our setup to evaluate the model-based behavior of MuZero on a variation of the classic Mountain Car task.

## 1 Introduction

Deep reinforcement learning (RL) has seen great success [13, 17, 6, 2], but it's no coincidence that these successes are almost exclusively on simulated environments—where samples are cheap—as contemporary deep RL methods have notoriously poor sample complexity. Deep model-based RL is a promising direction to substantially improve sample-efficiency. With model-based RL, an estimate of the transition dynamics and reward function is formed and planning techniques are employed to derive a policy from these estimates. For long, this approach did not combine well with function approximation, especially deep neural networks, due to fundamental problems such as compounding errors when predicting multiple steps into the future [20]. Recently, however, a number of papers have come out reporting success with deep model-based RL [15, 7–9, 4], including state-of-the-art performance on the Atari benchmark [16].

These recent successes bring to the forefront some interesting research questions, such as, *"What strategies are employed to mitigate the compounding error issue?"*, *"What are the relative strengths and weaknesses of the various deep model-based methods?"*, and, last but not least, *"How much room is there for further improvement?"*. Addressing such questions requires a clear notion of the target of model-based learning and a metric to measure progress along this target.

Currently, a common metric to evaluate model-based methods is single-task sample efficiency. However, this is a poor metric to measure progress on model-based learning for a number of reasons. First, it conflates improvements due to model-based learning with various other aspects, such as generalization and exploration. Second, it provides little insight into the relative performance

compared to an ideal model-based method. And last but not least, single-task sample-efficiency is arguably not the problem setting that best shows the relevance of model-based learning. The true sample-efficiency benefits manifest themselves when an agent can re-use its model for multiple tasks.

Inspired by work from neuroscience for detecting model-based behavior in humans and animals [5], we define a problem setup to identify model-based behavior in RL algorithms. Our setup is build around two tasks that differ from each other only in a small part of the state-space, but have very different optimal policies. Our *Local Change Adaptation (LoCA) regret* measures how quickly a method adapts its policy after the environment is changed from the first task to the second. Our setup is designed such that it can be combined with tasks of various complexity and can be used to evaluate any RL algorithm, regardless of what the method does internally. The LoCA regret can uniquely identify model-based behavior even when a method uses a poor representation or uses a poor choice of hyper-parameters that cause it to learn slowly. In addition, the LoCA regret gives a quantitative estimate of how close a method is to ideal model-based behavior.

## 2  Notation and Background

An RL problem can be modeled as a Markov Decision Process $M = \langle \mathcal{S}, \mathcal{A}, P, R, \gamma \rangle$, where $\mathcal{S}$ denotes the set of states, $\mathcal{A}$ the set of actions, $P$ the transition probability function $P : \mathcal{S} \times \mathcal{A} \times \mathcal{S} \to [0, 1]$, $R$ the reward function $R : \mathcal{S} \times \mathcal{A} \times \mathcal{S} \to \mathbb{R}$, and $\gamma$ the discount factor. At each time step $t$, the agent observes state $s_t \in \mathcal{S}$ and takes action $a_t \in \mathcal{A}$, after which it observes the next state $s_{t+1}$, drawn from the transition probability distribution $P(s_t, a_t, \cdot)$, and reward $r_t = R(s_t, a_t, s_{t+1})$. A *terminal state* is a state that, once entered, terminates the interaction with the environment; mathematically, it can be interpreted as an absorbing state that transitions only to itself with a corresponding reward of 0.

The behavior of an agent can be described as a policy $\pi$, which, at time step $t$, takes as input the history of states, actions, and rewards, $s_0, a_0, r_0, s_1, a_1, \ldots r_{t-1}, s_t$, and outputs a distribution over actions, in accordance to which action $a_t$ is selected. If action $a_t$ only depends on the current state $s_t$, we will call the policy a *stationary* policy. For each MDP there exists a stationary optimal policy $\pi^*$ that maximizes the discounted sum of rewards, $\sum_{i=0}^{\infty} \gamma^i r_i$, and the goal of an RL algorithm is to find this optimal policy after interacting with the environment for a sufficient amount of time. Note that any learning algorithm, by its nature, implicitly implements a non-stationary policy, because the actions it takes in any particular state will change over time in response to what it learns about the environment. The specific way an agent changes how it takes actions in response to newly observed information can be used to classify the agent's behavior as, for example, model-free or model-based behavior.

Traditional (tabular) model-based RL methods estimate $P$ and $R$ from samples and then use a planning routine like value iteration to derive a policy from these estimates [3, 11]. Such methods typically include strong exploration strategies, giving them a huge performance edge over model-free RL methods in terms of single-task sample efficiency, especially for sparse-reward domains. In most domains of interest, however, the state space is either continuous or can only be observed indirectly via a (high-dimension) observation vector that is correlated, but does not necessarily disambiguate the underlying state. On such domains, these traditional model-based methods cannot be applied.

To deal with these more complex domains, deep model-based RL methods often learn some embedding-vector from observations and learn transition models based on embeddings. In contrast to their traditional counterparts, deep model-based methods typically use naive exploration strategies, as effective exploration is still a very challenging problem for domains with state spaces more sophisticated than finite tabular ones. We discuss the various approaches to deep model-based RL in more detail in Section 4. First, we dive into the main topic of this paper: measuring model-based behavior.

## 3  Measuring Model-Based Behavior

In neuroscience and related fields, a well-established hypothesis about decision making in the human brain is that there are multiple, distinct systems. There is a habitual, or model-free system that evaluates actions retrospectively by assigning credit to actions leading up to a reward, much like what happens in temporal-difference learning. There is also a more deliberative, model-based system that evaluates actions prospectively, by directly assessing available future possibilities. These

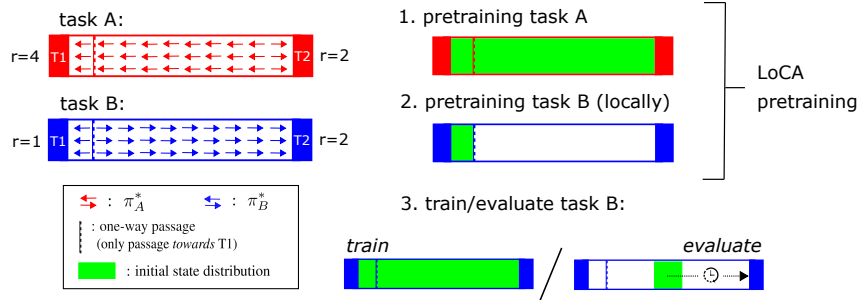

Figure 1: Experimental setup for evaluating model-based behavior. Task A and task B have the same transition dynamics, but a reward function that is locally different. $\gamma$ should be close to 1, to ensure that for most of the state-space, an optimal agent moves toward the high-reward terminal. An experiment consists of first pretraining a method on task A, followed by local pretraining around T1 of task B. After pretraining, the agent is trained on the full environment of task B. At periodic intervals, the agent is evaluated by determining if it can reach T2 within a certain number of time steps, starting from an initial state distribution that is roughly in the middle. The additional training a method requires to pass this evaluation test determines the size of the LoCA regret.

different modes of decision-making lead to distinct behavioral patterns that can be identified using appropriately set-up experiments, such as the two-step decision-making task introduced in [5].

Artificial agents are not necessarily bound to these two modes of learning. Variations of these modes can be developed along different dimensions to the point that using a binary label to classify such methods loses its usefulness [18]. Moreover, similar behavior may be generated using very different internal processes [21]. Nevertheless, it is useful to consider what behavioral characteristics an ideal model-based agent would exhibit, given the neuroscience interpretation of a model-based system.[1] Such an agent is able to make accurate predictions about future observations and rewards for any hypothetical sequence of actions. And it can use this capability to reason about the consequences of all possible future action-sequences and deduce from this the optimal action to take at the current time. Each new observation updates its internal model of the world and hence each time it has to take a new action, this reasoning process is repeated such that the agent always takes the best action using its most up-to-date model of the world.

The speed at which newly observed information results in policy changes throughout the state-space is one of key behavioral characteristic of model-based learning that sets it apart from other forms of learning. And therefore our experimental setup that is introduced below has been build around measuring this speed in a way that removes confounding effects as much as possible.

## 3.1 Experimental Setup

The behavior we want to evaluate is how quickly newly observed information influences the policy throughout the state space. We formalize this as follows: consider two MDPs with shared state and action space: $M_A = \langle \mathcal{S}, \mathcal{A}, P_A, R_A, \gamma \rangle$ and $M_B = \langle \mathcal{S}, \mathcal{A}, P_B, R_B, \gamma \rangle$, where $P_A$ and $R_A$ differ from $P_B$ and $R_B$ only locally. That is, only a small number of state-action pairs have different transition dynamics and/or reward. Let $\pi_A^*$ be the optimal policy for task A, and $\pi_B^*$ be the optimal policy task B. Note that a local difference in dynamics/reward can result in a global difference between $\pi_A^*$ and $\pi_B^*$. Furthermore, let $\pi_A \approx \pi_A^*$ be the policy after training a method on task A; and $\pi_B \approx \pi_B^*$ after training that method on task B. We want to measure how quickly an agent that is trained on task A changes its policy from $\pi_A$ to $\pi_B$ after it is placed in task B and has observed the local difference in dynamics/reward.

In the specific experimental setup we propose, task A and task B contain two terminal states, T1 and T2, at opposite sides of the state space (see Figure 1). The transition dynamics of the two tasks are the same ($P_A = P_B$), but there is a local difference in the reward function. Specifically, the reward for reaching T1 is 4 for task A and 1 for task B. The reward for reaching T2 is 2 for both tasks and

the reward is 0 everywhere else. There is a one-way passage close to T1 that guarantees that when an agent is initialized close to T1, it will never observe anything beyond the local environment of T1, no matter the actions it takes.

An experiment consists of three phases. In the first phase, the agent is trained on task A until all of its internal representations have converged to their final values (whether this is a model, a value function, a policy estimate, or a combination of these). In the second phase, the agent is placed on task B and trained on the local environment of T1, by using an initial state distribution beyond the one-way passage. In the third phase the agent is trained on task B using an initial state distribution that enables the agent to explore the full state space. The first two phases are pretraining phases. We collectively refer to these two phases as *LoCA pretraining*. Only the behavior shown during the third phase is used to evaluate the agent.

The agent is evaluated by stopping training during phase three at regular intervals and running the agent for $n$ evaluation episodes on task B, using an initial state distribution approximately in the middle of T1 and T2. Under policy $\pi_A^*$, trajectories starting from this distribution end up in T1; by contrast, under policy $\pi_B^*$, they terminate in T2. To evaluate an agent, we measure the fraction of evaluation episodes that the agent ends up in T2 within a predefined number of time steps. We refer to this fraction as the *top-terminal fraction*. Our main metric, the LoCA regret, is based on this fraction, as we will show below.

## 3.2   Evaluation of Experimental Setup

In this section, we analyze in more detail the usefulness of the setup using a simple tabular navigation domain (Figure 2) for which strong model-based methods can easily be implemented. We used a discount factor $\gamma$ of 0.97, which ensures that for most the state-space an optimal agent moves to the high-reward terminal.

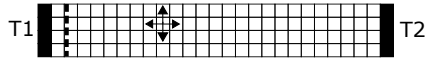

Figure 2: Gridworld task ($25 \times 4$).

We compare the performance of the model-free method Sarsa($\lambda$), using $\lambda = 0.95$ and 'dutch traces' [22], with a model-based method with and without pretraining. The model-based method, MB-VI, simply learns a (tabular) model of the transition dynamics and reward and uses value iteration to derive a policy from the model estimate. In addition, we evaluate both methods on a variation of the task, where we expand the state space with a random variable that does not affect the dynamics or reward, artificially increasing the size of the state space by a factor $S_{multp}$. This allows us to study the effect of representation learning, even though the task is strictly tabular. In particular, if method 1 uses a state-space with additional random features and method 2 uses a state-space without such features, then method 2 can be viewed as having a representation-learning module compared to method 1. Because if method 2 was given the same state-space as method 1, but would also have a representation-learning module that learns to ignore the irrelevant random features during LoCA pretraining, the performance would be the same (in Section 3.3, we provide further motivation for why this interpretation of representation learning is useful).

The results shown in Figure 3 demonstrate that with LoCA pretraining model-based methods have a top-terminal fraction of 1 from the very beginning (i.e., the start of phase 3 in Figure 1), whereas model-free methods require additional training time to reach a top-terminal fraction of 1. Furthermore, note that Sarsa($\lambda$) and MB-VI, $S_{mult} = 5$ have a similar performance when no pretraining is used, despite being very different methods. In this case, Sarsa($\lambda$) can be viewed as operating on the same state-space as MB-VI, $S_{mult} = 5$, but using a representation-learning module that has learned to ignore the irrelevant features, whereas MB-VI, $S_{mult} = 5$ does not have this capability, but uses model-based updates. While these lead to similar performance when no pretraining is used, LoCA pretraining can uniquely identify the model-based method. Finally, comparing Sarsa($\lambda$) with Sarsa($\lambda$), $S_{mult} = 2$ shows that the effect of representation learning does not vanish with LoCA pretraining in the case of model-free methods, in contrast to model-based methods.

We can summarize the behavior succinctly by measuring the regret with respect to an optimal policy:

$$\text{regret} \quad = \sum_{i=0}^{\infty}(1 - \mathfrak{f}_i) \cdot \Delta_{train}$$

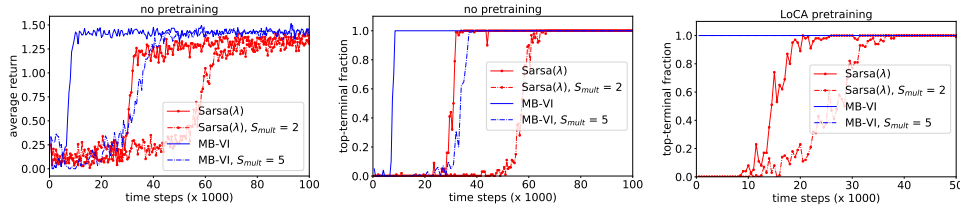

Figure 3: Performance on task B for model-based and model-free methods. $S_{mult}$ indicates the state-space multiplier (to mimic poor representation); $\alpha_{mult}$ indicates the step-size multiplier (to mimic poor hyper-parameters). **(left)** traditional average return metric and no pre-training. **(middle)** top-terminal fraction with no pre-training. **(right)** top-terminal fraction with LoCA pretraining (the results for MB-VI and MB-VI, $S_{mult} = 5$ overlap exactly).

where $\mathfrak{f}_i$ the $i$-th evaluation of the top-terminal fraction and $\Delta_{train}$ the number of training time steps between consecutive evaluations.[2] If this regret is used in combination with LoCA pretraining, we refer to this regret as the *LoCA regret*.

To study the experimental setup in more detail, we compute the default regret as well as the LoCA regret for the methods shown in Figure 3 , as well as various additional ones. In particular, we also show the regret for a method, indicated by MB-SU, that uses the same model-based learning as MB-VI, but instead of doing value iteration at each time step, it performs a single value update for the current state. This helps us understand the behavior of a model-based method with a low-quality planning routine. Furthermore, to mimic the use of poor hyperparameters, we show results of the various methods when the step-size parameter $\alpha$ is 10 times as small( $\alpha_{multp} = 0.1$) as the default value used. For Sarsa($\lambda$), the step-size controls the size of the model-free updates; for MB-VI and MB-SU the step-size controls the size of the model updates.

The results from Table 1 reveal some key insights about our setup:

1. *A method that learns an accurate environment model in the end and has a strong planning routine obtains 0 LoCA regret, even if it learns the model slowly because it includes irrelevant features or uses sub-optimal model-learning hyperparameters.*

2. *A method that does not learn a model or learns a model but uses poor planning can be affected by the quality of representation; nevertheless, a great representation cannot make up for the lack of a model and/or poor planning.*

Based on these observations, we argue that our setup and regret metric are useful for measuring progress on model-based learning. While initial progress can be made by simply focusing on finding a good representation or implementing techniques to boost model-free learning (for example, using eligiblity traces), at some point, to make further progress, the agent has to start learning a model of the environment and implement a strong planning routine. Furthermore, the better the model and the stronger the planning, the less the generalization behavior of a method matters. Note that representation learning can still be a critical component for model-based methods indirectly, because it may affect the quality of the overall model that can be learned.

While the LoCA regret is our main performance metric for measuring model-based behavior, it can be useful to consider the gain achieved from LoCA pretraining. The last column in Table 1 shows the gain relative to a Q-learning baseline, computed as follows:

$$\text{gain} = \frac{\text{default regret}}{\text{LoCA regret}} \qquad ; \qquad \text{relative gain} = \frac{\text{gain method}}{\text{gain baseline}} .$$

A gain relative to a model-free method like Q-learning that is significantly larger than 1 suggests that a method learns something useful during pretraining that can be transferred between task A and task B. Note that in order to get an unbiased gain, methods should be initialized similarly, as initialization (f.e. optimistic versus pessimistic initialization) affects the regret without pretraining.

Table 1: Regret ($\times 1000$) *(and standard error)* of various methods with and without LoCA pretraining, averaged over 10 runs.

| method | default regret | | LoCA regret | | relative gain |
|---|---|---|---|---|---|
| Sarsa(0.95) | 30.09 | *(0.22)* | 14.92 | *(0.23)* | 1.17 |
| Sarsa(0.95), $\alpha_{multp} = 0.1$ | 283.95 | *(0.88)* | 134.82 | *(1.71)* | 1.22 |
| Sarsa(0.95), $S_{multp} = 2$ | 56.80 | *(0.37)* | 26.30 | *(0.36)* | 1.25 |
| Q-learning | 68.25 | *(0.15)* | 39.52 | *(0.18)* | 1.00 |
| MB-VI | 7.29 | *(0.08)* | 0.00 | *(0.00)* | $\infty$ |
| MB-VI, $\alpha_{multp} = 0.1$ | 78.79 | *(0.26)* | 0.00 | *(0.00)* | $\infty$ |
| MB-VI, $S_{multp} = 5$ | 32.98 | *(0.19)* | 0.00 | *(0.00)* | $\infty$ |
| MB-SU | 10.25 | *(0.06)* | 1.72 | *(0.06)* | 3.45 |
| MB-SU, $\alpha_{multp} = 0.1$ | 84.83 | *(0.29)* | 1.87 | *(0.06)* | 26.27 |
| MB-SU, $S_{multp} = 10$ | 85.61 | *(0.22)* | 15.67 | *(0.10)* | 3.16 |

## 3.3 Further Considerations

Many existing tasks can be adapted to follow the experimental setup outlined above. For example, we combine the setup with the classical Mountain Car task in Section 5, by adding an additional terminal state. The main requirement is that the (modified) task contains two terminal states and some notion of a one-way passage. Of course, not every existing task can be easily modified in this way without fundamentally changing the nature of the task itself, but that is not the goal. The goal is to accurately evaluate the behavior of an RL method in a controlled setting, much like in neuroscience cleverly constructed tasks are used to disentangle the modes of learning in humans or rodents. One great advantage of our particular setup is that it can give an indication of how far the behavior of a method deviates from ideal model-based behavior without having access to an ideal model-based method.

For the LoCA regret to be measured, a method should be able to find a near-optimal policy for the considered domain with enough training data. We argue that this is not a strong limitation; if methods are not able to achieve good performance on a domain in the first place, the speed of adaptation is likely not the main concern. Related to this, a domain does not necessarily have to be very complicated, in order to reveal interesting results, as the central question is not whether or not a method can solve the domain; the central question is the speed at which a method adapts to a local change in the domain.

The way we evaluate the effect of representation learning in Section 3.2 may seem limited. In particular, in our tabular setup a poor representation can simply be compensated for with additional training data, which is not necessarily true for all representations. However, as mentioned above, a prerequisite for computing the LoCA regret is that a method can find a near-optimal policy given enough training data. And under this condition, the difference between a poor and a good representation expresses itself only through data efficiency, also in non-tabular settings, hence the relevance of these results.

## 4 Taxonomy of Deep Model-Based Methods

In this section, we review the landscape of deep model-based methods for reinforcement learning. Deep model-based methods typically learn the forward model of the world which includes a transition function and reward function. While designing a model-based RL algorithm, there are a number of important design choices to make, such as, *"what type of model to learn?"* and *"how to use the learned model?"*. We discuss the key differences among current deep model-based methods below.

**What type of model to learn?** The state transition model could either model the entire transition probability distribution (stochastic model) or just learn to predict the expected next state (expectation model). While stochastic models are richer than expectation models, they are also difficult to learn. While methods like MuZero [16], Imagination Augmented Agents (I2A) [15], TreeQN and ATreeC [7] learn expectation models, World Models [8], PLANET [9], DREAMER [10] and PETS family [4, 23, 14] learn stochastic models of the world. All the above mentioned models except PETS also learn the reward function. PETS assumes access to the true reward function which makes it limited when compared to other methods.

**How to use the learned model?** There are three major ways in which the learned model is used by the model-based algorithms. One can use the learned model of the world *to search* for an action sequence that would give the highest return. This is the common application of the model and has been done by MuZero, TreeQN, ATreeC, PLANET, and PETS to name a few algorithms. The other option is to use the model in *Dyna-style* [18] where the model is used to generate samples which will be used to improve the model-free components of the algorithm. World Models and DREAMER follow this strategy. Third way of using the model is as a representation enhancement for the model-free RL algorithms. Examples for this application includes I2A.

**How is the model trained?** Model-based methods can be categorized into three types based on the objective function that the model is trying to optimize for. Algorithms like World Models, I2A, PLANET, DREAMER, PETS train the model with forward prediction error which aims to learn models that are as accurate as possible in predicting the future states. On the other hand, the models can also be trained directly with the final task loss in which case models are expected to be useful for the task in hand. This includes MuZero, TreeQN, and ATreeC which all learns the model that minimized the reinforcement learning objective. Other distinction in training the models is whether they are trained end-to-end or in phases. Most of the models trained with forward prediction error are trained in phases where the model is first trained with collected trajectories and then the trained model is used for planning separately. World Model, and I2A followed this strategy. However, other algorithms like MuZero, ATreeC, TreeQN, PLANET, DREAMER and PETS train the model simultaneously while learning the end task.

**What is the input to the model?** The forward models can be learnt either directly from the pixel-level observation space or from a learnt latent embedding space. While algorithms like I2A and PETS try to predict the next state directly in the observation space, it is not feasible for complex environments. It only works for either simple environments or environments with low-dimensional observations. On the other hand, algorithms like MuZero, World Models, TreeQN, ATreeC, PLANET, and DREAMER operate in the latent embedding space.

**What environments can be modeled?** Almost all the model-based RL algorithms we have discussed in this section has been tested only in fully observable or almost fully observable (like Atari) environments. Also they are all tested only on deterministic environments where learning an accurate model of the world is relatively easier. World Models is an exception since the authors showed some results with stochastic environments and the model itself has some inductive bias which exploits the stochasticity of the environment. Except for MuZero, PLANET, and DREAMER, all these algorithms have been demonstrated to work only in dense reward setting while most of the interesting problems are sparse reward in nature.

## 5 Experiments

First, we perform additional experiments on the tabular navigation task shown in Figure 2. In particular, we explore how an on-policy model affects model-based behavior. After this, we evaluate the behavior of MuZero and PLANET on a variation of the Mountain Car task.[3]

### 5.1 Tabular On-policy Model

An on-policy model is a (transition) model whose predictions are conditioned on one particular policy, typically the behavior policy. An example of that is a simple $n$-step model that predicts, for each state-action pair, what the state observed $n$ steps in the future will be; the corresponding reward model predicts the discounted sum of reward accrued over the next $n$ time steps.

Table 2 shows the regret with and without pretraining for a 1-step, 2-step and 5-step model (for further details, see the supplementary material). Note that a 1-step model is the only model that

Table 2: Regret ($\times 1000$) with and without LoCA pretraining .

| method | default regret | | LoCA regret | | relative gain |
|---|---|---|---|---|---|
| 1-step model | 42.98 | *(0.17)* | 1.68 | *(0.06)* | 14.81 |
| 2-step model | 40.25 | *(0.27)* | 24.87 | *(0.77)* | 0.94 |
| 5-step model | 34.51 | *(0.39)* | 34.77 | *(1.83)* | 0.57 |

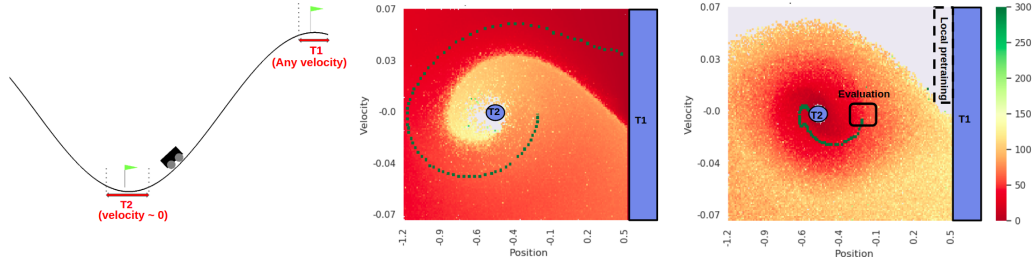

Figure 4: Mountain Car task with two terminal states **(left)**. The distance to terminal state T1 **(middle)** and T2 **(right)**, after training Sarsa($\lambda$) on task A and task B, respectively (grey means the agent did not reach the designated terminal state). An example trajectory is plotted for both cases starting from the same initial state.

is not on-policy. What these results show is that while adding on-policy elements to a method can help with single-task sample-efficiency, they can hurt the model-based behavior of a method.

## 5.2 MuZero (and PLANET)

Next, we evaluate two deep model-based methods on a variation of the Mountain Car task, the classic benchmark task in which an underpowered cart has to move up a hill. In our variation of the task, the existing terminal state at the top of the hill corresponds with T1; we added an additional terminal state to the domain, T2, that corresponds with the cart being at the bottom of the hill with a velocity close to 0. We did not explicitly encode a one-way passage. Instead, close to T1 (for position $> 0.4$) we map each action index to the action pushing the cart to the right. Effectively, this creates a local area close to T1 from which the cart cannot escape (i.e., the cart will inevitably hit T1 within a few time steps). For a detailed view of the state-space, including the evaluation and phase 2 (local pretraining) initial-state distribution, see Figure 4.

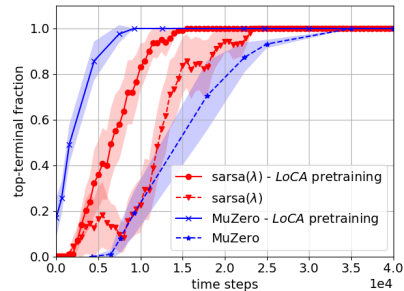

Figure 5: Performance with and without LoCA pretraining of Sarsa($\lambda$) and MuZero ($K = 3, \epsilon = 0.2$).

The model-based methods that we evaluate are MuZero, using the implementation provided by [12], and PLANET, using the implementation provided by [1]. Out of all the deep model-based methods discussed in the previous section, MuZero is arguably the most impressive one, as it can deal well with high-dimensional, long-horizon domains, as demonstrated by its state-of-the-art performance on the Atari benchmark. PLANET is interesting because: 1) it has also been shown to work on sparse-reward domains, and 2) in contrast to MuZero (and DREAMER), it does not contain obvious on-policy elements such as bootstrapping from a value function.

Because PLANET requires continuous actions, we tested it on a version of Mountain Car that uses continuous actions, but still has a similar horizon (i.e., the number of steps it takes to go from a state in the evaluation area to T1 or T2 is similar). PLANET was unable to solve the task—a requirement for determining the LoCA regret—and therefore, we only show the results for MuZero below. The most likely reason for PLANET's poor performance is the long horizon of the Mountain Car task (80-120 steps), which is considerably longer than the horizon of the spare-reward tasks PLANET has been shown to work on (f.e., Reacher has a horizon of 10-15 steps).

We compare the performance of MuZero with that of Sarsa($\lambda$) with $\lambda = 0.9$. For Sarsa($\lambda$), we used the common linear representation based on tile-coding [19]. For MuZero, we varied two important hyperparamaters: $K$, the number of steps the model is unrolled recurrently during training ($num\_unroll\_steps$ in the code), and $\epsilon$ ($root\_exploration\_fraction$ in the code), which controls how much noise is added to the MCTS prior of the root node's actions. High values of $\epsilon$ cause MCTS to explore a range of actions for the root node, rather than focusing primarily on the on-policy action. We used 50 MCTS simulations per time step ($num\_simulations$ in the code), which is the

default value used by MuZero. We also tried running 100 simulations per time step, but this did not substantially change the results.

Table 3 shows the results for several $\epsilon$-values and two values of $K$. We explored $\epsilon$-values in the range from $0.1 - 0.9$, finding that $\epsilon = 0.2$ provides the best LoCA regret (Figure 5 shows the corresponding

Table 3: Regret ($\times 1000$) with and without LoCA pretraining

| method | default regret | | LoCA regret | | relative gain |
|---|---|---|---|---|---|
| Sarsa($\lambda$) | 12.55 | *(1.84)* | 7.30 | *(1.05)* | 1.0 |
| MuZero, $K = 3, \epsilon = 0.1$ | 20.86 | *(0.85)* | 5.93 | *(0.58)* | 2.01 |
| MuZero, $K = 3, \epsilon = 0.2$ | 15.21 | *(1.23)* | 2.24 | *(0.41)* | 3.25 |
| MuZero, $K = 3, \epsilon = 0.4$ | 19.42 | *(0.96)* | 4.35 | *(0.19)* | 2.59 |
| MuZero, $K = 5, \epsilon = 0.1$ | 14.45 | *(0.24)* | 6.12 | *(0.85)* | 1.37 |
| MuZero, $K = 5, \epsilon = 0.2$ | 12.16 | *(0.84)* | 2.46 | *(0.38)* | 2.87 |
| MuZero, $K = 5, \epsilon = 0.4$ | 13.57 | *(0.78)* | 4.66 | *(0.31)* | 1.66 |

curve). For $\epsilon$ equal to 0.6 and higher, MuZero no longer converged to the correct policy. Increasing $K$ from 3 to 5 improved the default regret, but it did not improve the LoCA regret.

## 6 Discussion and Future Work

Overall, our results show that while MuZero substantially outperforms Sarsa($\lambda$) in terms of the LOCA regret, it is still a fair amount away from ideal model-based behavior. This is a surprising result, given the simplicity of the Mountain Car task and the fact that MuZero has been shown to achieve state-of-the-art performance on Atari in terms of single-task sample efficiency. More generally, our results highlight the need for alternative metrics, such as the LoCA regret, for measuring progress on model-based learning. More experiments are needed to understand exactly why MuZero is unable to achieve 0 LoCA regret. But the tabular results shown in Table 2 give a potential explanation: adding on-policy elements to a model may help improve single-task sample efficiency, but this appears to hurt a method's ability to quickly adapt to local changes substantially.

In future work, we plan to perform an extensive empirical comparison of the various deep model-based methods with respect to the LoCA regret. Besides considering various methods/techniques, we would like to consider various task-dimensions that can make a problem more or less challenging from a model-based perspective. For example, expectation models, such as the one used by MuZero, can result in large compounding errors when the domain is stochastic. So it would be interesting to consider how MuZero's performance is affected by stochasticity in the domain.

## 7 Conclusion

We introduced a setup and corresponding metric to measure how quickly an RL agent adapts to a local change in the environment. Adapting quickly to small changes in the environment is a key feature of model-based behavior and, in and of itself, a skill that is essential for many practical applications. Our results show that MuZero is a fair amount away from ideal model-based behavior, demonstrating that achieving great performance in terms of single-task sample-efficiency does not guarantee great performance in terms of the ability to quickly adapt to changes. More experiments are needed to understand what prevents MuZero from achieving perfect model-based behavior, but some of our results suggest that adding on-policy elements to a method, while advantageous in the context of single-task sample efficiency, can seriously impede a method's ability to quickly adapt.

## Broader Impact

The setup and metric introduced in this paper helps speed up progress towards RL methods that can quickly adapt to changes in the environment. Adaptability with respect to environmental changes is a key requirement for many real-world applications, but it is something that current state-of-the-art RL methods struggle with. As such, this work can be viewed as a step towards pushing RL from simulated, stationary environments towards real-world applications.

The rise of automated decision-making systems can have many societal consequences, both positive and negative. Certain types of jobs may be replaced by automated systems, for example, various

types of help-desk services. Another aspect to consider is bias: relying more on automated decision-making system could potentially avoid biases present in human decision-making, for example, in legal or healthcare services; on the other hand, there is a risk of introducing new biases related to how the system is trained, algorithmic biases or the way the system is used. Therefore, a valuable avenue for future work to increase the chance of positive impact is to study how biases in automated decision-making systems can be detected and how they can be mitigated.

## Acknowledgments and Disclosure of Funding

We would like to acknowledge Compute Canada and Calcul Quebec for providing computing resources used in this work. SC is supported by a Canada CIFAR AI Chair and an NSERC Discovery Grant.

## Footnotes

[1]In the context of this paper, we ignore the computational aspects of various approaches, even though in neuroscience, the time taken to make a decision is sometimes used as a feature to identify the mode of learning.

[2]To get a finite regret, a method needs to solve the task eventually; if this is not the case, a finite-horizon version of the regret can be used.

[3]The code of the experiments is available at `https://github.com/chandar-lab/LoCA`. Additional implementation details can be found in the supplementary material.

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
