[Supplementary Material]

# The LoCA Regret: A Consistent Metric to Evaluate Model-Based Behavior in Reinforcement Learning
## — *Supplementary Material* —

## A    Tabular Experiments

Here, we discuss some additional settings for the tabular experiments. Full code is available at `https://github.com/chandar-lab/LoCA`. For all tabular experiments, we used $\epsilon$-greedy exploration with $\epsilon = 0.1$. Furthermore, during pretraining and training, we used a maximum episode-length of $100$. For evaluation, we set $\epsilon = 0$, and ran 10 evaluation episodes. For those 10 episodes, we counted the fraction of episodes that the agent reached terminal 2 within $40$ steps. All experiments are repeated over 10 independent runs and averaged.

We used a fixed step-size $\alpha$ for all tabular experiments. For the model-based methods (MB-VI, MB-SU and $n$-step model), $\alpha$ is used in the update of the (tabular) transition model and reward function. For Sarsa($\lambda$) and Q-learning, $\alpha$ is used to update the action-value function. For MB-VI and MB-SU we used $\alpha = 0.2$. Because the environment is deterministic, higher values of the step-size would have been possible here. However, we opted for not making the step-size too high, because we are ultimately interested in the function approximation scenario where large step-sizes are not an option. For Sarsa(0.95), we used $\alpha = 0.05$, because for higher step-size values performance became unstable. The reason for this is that Sarsa(0.95), in contrast to MB-VI and MB-SU, is a multi-step method. Therefore, there is stochasticity in the update target even in deterministic environments due to exploration of the behavior policy. For the same reason, we used $\alpha = 0.04$ for the $n$-step models: for higher values the performance of the 5-step model became unstable. While for smaller values of $n$, a higher step-size could be used, we used the same step-size for all $n$-step models to ease comparison.

The initialization of a method does not affect the LoCA regret, because these effects are removed by the first pretraining phase (pretraining on task A until convergence). For the default regret, however, which does not use LoCA pretraining, the initialization of a method can affect performance. And because the relative gain is based among others on the default regret, the initialization can affect the relative gain as well. To remove bias from initialization effects from the default regret (and hence, the relative gain), there are two options: 1) ensuring that all considered methods are initialized similarly; 2) pretraining on a unrelated task from which no useful information can be transferred. For the tabular methods, we chose the former option. All methods used optimistic initialization. For Sarsa($\lambda$) and Q-learning, the action-value function was initialized at $4$. For the model-based methods, the initial transition model predicts a terminal state for each state-action pair; the initial reward model predicts a reward of $4$ for each state-action pair.

### A.1    On-Policy Model-Based Method

The pseudocode of the tabular, on-policy method used in Section 5.1 is shown in Algorithm 1. The algorithm maintains estimates $\hat{P}^n$ and $\hat{R}^n$ of the $n$-step transition model and n-step reward function:

$$
\begin{aligned}
P_\pi^n(z|s,a) & := \mathbb{P}(S_{t+n} = z | S_t = s, A_t = a, \pi\} \\
R_\pi^n(s,a) & := \mathbb{E}\{R_t + \gamma R_{t+1} + \cdots + \gamma^{n-1} R_{t+n-1} | S_t = s, A_t = a, \pi\}
\end{aligned}
$$

These estimates are updated at the end of the episode, using the data gathered during the episode. If $t + n$ falls beyond the end of an episode, then $P^n(\cdot|S_t, A_t)$ is updated using the terminal state (and rewards are 0 after the end of an episode). The algorithm also maintains an estimate of the value

function, $\hat{V}$. We use a conservative update strategy for updating the state-value function, by only updating the value of the current state. The algorithm does not store a Q-value function, but computes estimates of the Q-values for the current state on-the-fly using its estimates of $\hat{P}^n$, $\hat{R}^n$ and $\hat{V}$.

---

**Algorithm 1** tabular n-step model-based method

---

For all $s$: $\quad \hat{V}(s) \leftarrow 0$
For all $s, a$: $\quad \hat{R}^n(s, a) \leftarrow 0$
For all $s, a, z$: $\quad \hat{P}^n(z|s, a) \leftarrow 0$
Loop (over episodes):
$\quad$ $t \leftarrow 0$
$\quad$ $episode\_samples \leftarrow \emptyset$
$\quad$ obtain initial state $S$
$\quad$ While $S$ is not terminal, do:
$\quad\quad$ For all actions $a$:
$\quad\quad\quad$ $Q(a) \leftarrow \hat{R}^n(S, a) + \gamma^n \sum_z \hat{P}^n(z|S, a) \cdot \hat{V}(z)$
$\quad\quad$ $\hat{V}(S) \leftarrow (1 - \alpha) \cdot \hat{V}(S) + \alpha \cdot \max_a Q(a)$
$\quad\quad$ select action A based on $Q(a)$
$\quad\quad$ take action A, observe next state $S'$ and reward $R$
$\quad\quad$ add $(S, A, R)_t$ to $episode\_samples$
$\quad\quad$ $S \leftarrow S'$
$\quad\quad$ $t \leftarrow t + 1$
$\quad$ For all $(S_t, A_t)$ in $episode\_samples$
$\quad\quad$ update $P^n(\cdot|S_t, A_t)$ using $S_{t+n}$
$\quad\quad$ update $R^n(S_t, A_t)$ using $R_{t+1} + \gamma R_{t+2} + \ldots + \gamma^{n-1} R_{t+n}$

---

## B  Mountain Car Experiments

In this section, we describe the implementation details of Mountain Car experiments.

### B.1  Environment

We used a modified version of the classic mountain car environment for our experiments. In particular, we added another terminal state, $T_2$, that corresponds to the car being at the bottom of the hill with a velocity close to zero ($[(x - 0.52)^2 + (10v)^2] < R^2$). In our experiments, we chose $R = 0.07$. Since we found MuZero struggling to reach $T_2$, we decided to sample the initial state for both MuZero and Sarsa($\lambda$) from a mixture of two uniform distributions. With the probability $p$, the initial state is sampled from the whole state space ($x \sim [-1.2, 0.5]$ and $v \sim [-0.07, 0.07]$) and with probability $1 - p$, it is sampled from a smaller box around $T_2$ ($x \sim [-1, 0]$ and $v \sim [-0.03, 0.03]$). We set $p = 0.5$ for our experiments. Moreover, during evaluation phase, the initial state is sampled from an area which has almost same distance to each terminal ($x \sim [-0.2, -0.15]$ and $v \sim [-0.005, 0.005]$).

### B.2  Sarsa($\lambda$)

We used the common linear representation based on tile-coding, which uses 10 overlapping grid-tilings, each consisting of $10 \times 10$ tiles (for further details, see [2]). Furthermore, we used $\lambda = 0.9$ and a learning rate $\alpha = 0.05$. Also, we used $\epsilon$-greedy as the exploration strategy. During the first pretraining phase (pretraining on task A until convergence), $\epsilon$ has an initial value of 1 that exponentially decays to 0.01. Note that this $\epsilon$-decay is simply used to speed up the pretraining phase; it does not affect the LoCa regret. During the second pretraining phase (local pretraining on task B) and the training phase, we used a fixed $\epsilon$ of 0.1. And during evaluation, we used $\epsilon = 0$. Finally, we let the Sarsa($\lambda$) agent to be trained for 200 $k$ steps, 5 $k$ steps and 40 $k$ steps during phase 1, phase 2 and phase 3, respectively.

### B.3  MuZero

For our mountain car experiments, we used the same hyperparameters for MuZero as the ones used in [1], except those that we mention here. We let the MuZero agent to be trained for 200 $k$ steps, 5 $k$ steps and 40 $k$ steps during phase 1, phase 2 and phase 3, respectively.

### B.3.1 Network architectures

The input to the representation network is a two dimensional state of the environment. The size output of the representation network is 8 which will be used as the input to the prediction network. The input to the dynamics network is the hidden state produced by the representation network concatenated with a one-hot representation of the action taken. Dynamics network and prediction network consist of two fully-connected layers followed by the ReLU activation functions. We used 6 values from -1 to +4 for reward support and value support which are the output of reward prediction network and value prediction network, respectively. Furthermore, similar to $MuZero\ Reanalyze$ [1], we used a target network to have more stable and faster training.

Figure 1: Networks

## B.4 Default Regret

As mentioned in Section A, the initialization of a method can affect the default regret and, consequently, the relative gain. Furthermore, because Sarsa($\lambda$) and MuZero have very different internal representations, it is hard to guarantee that they are initialized in a fair way that does not give an implicit advantage to either one. For this reason, to compute the default regret, we pretrained both methods until convergence on a task from which no useful information can be transferred. Pretraining on the same task until convergence assures methods have the same starting point even if they use different internal representations and different initialization. The specific pretraining task we picked was a variation of task A with shuffled action-indices. Specifically, whereas action '0' normally pushes the cart to the left; in our shuffled version, action 0 is the no-op action. Similarly, action 1—normally the no-op action—pushes the cart to the right, and action 2—normally pushing the cart to the right—pushes the cart to the left. Note that no useful transition information can be learned from this form of pretraining, in contrast to LoCA pretraining.