[Reviews · NeurIPS 2020]

Review 1

Summary and Contributions: This paper aims to introduce a methodology for measuring the extent to which an RL agent uses model-based reasoning/learning. Inspired by methodologies in fields that study natural organisms, the main idea is to measure the speed at which the agent adapts its behavior in the face of a small, local change to dynamics that causes a large, global change in the value function. In principle an agent that make effective use of a model will adapt its policy without requiring much real-world training, once it has observed the change in dynamics. Results show that the LoCA regret can distinguish between clearly model-free and clearly model-based agents. The measure is also used to evaluate MuZero, indicating that it does display some model-based behavior, but also has room to improve its use of the model.

Strengths: - The paper raises an interesting and important issue. In algorithmic terms, the boundaries between model-free and model-based approaches are blurry. Agents that learn their model end-to-end, with only reward as feedback may simply be using their "model" as a complicated value function parameterization. This paper attempts to tease apart model-based and model-free *behavior* as distinct from the structures nominally used in the agent architecture. - The measure is based on behavioral science literature, which does lend it some additional credence. It essentially asks what would happen if someone were to evaluate this agent like they would a mouse; would they conclude that its decisions had a model-based component? - The utility of the measure is illustrated by applying it to MuZero, a recent and highly publicized agent architecture with model-based components. Though, simply from inspection, one can speculate that some of MuZero's design decisions would prevent it from obtaining all of the expected benefits of learning a model, it does seem meaningful to be able to quantify this gap and thus potentially determine whether improvements or alternatives are meaningfully closing it.

Weaknesses: - Though the LoCA regret is presented as a way to measure model-based behavior, there wasn't much justification that adaptation to local change is the right/only measure, other than its use in studies of biological organisms. I can imagine clearly model-based agents that might take a long time to adapt simply because of low step-sizes (e.g. to account for highly stochastic dynamics). I can also imagine model-free agents that might adapt very quickly (e.g. "The Brute" from Machado et al. JAIR 2018, which simply memorizes action sequences and their returns). The paper would be strengthened by a discussion of the limitations of this approach and exploration of distinctions it might not be suitable to make. - Though the LoCA regret is contrasted conceptually with single task sample complexity as an indicator of model use, the claims about the differences between the two is never quite evaluated. The paper would be strengthened by a clear example where LoCA is able to distinguish between methods where sample complexity improvement is caused by confounding factors and methods where sample complexity improvement is caused by effective planning.

Correctness: I am not aware of technical errors in this work.

Clarity: I found the paper to be clear and well-written. I feel that enough detail has been provided for an informed reader to reproduce these experiments.

Relation to Prior Work: I felt that the paper did a good job of surveying recent attempts to incorporate models in deep RL. On the one hand I am not sure why this discussion was limited to methods using deep neural networks. On the other, I certainly understand that this family of approaches has show the most scaling potential while also making the distinction between model-based and model-free agents the most ambiguous. I am not aware of existing alternatives to this paper's approach to this particular problem, so the lack of a direct comparison seems appropriate.

Reproducibility: Yes

Additional Feedback: After Response: Thank you to the authors for the response. My comments were addressed to my satisfaction. During the discussion phase concern was raised about the description of the MuZero experiments and parameterization. I agree that this is an issue and this part of the text needs to be re-written to make it clear precisely what parameters were varied and what impact this would have on the agent.


Review 2

Summary and Contributions: This paper develops a new evaluation test for RL agents, which claims to identify good model-based agents. The authors illustrate this test for some simple problems using several different agents. The idea is interesting and worth pursuing but the current results were not quite compelling.

Strengths: I think the idea of using specially designed tests for determining agent sophistication is an important direction for research. The authors have put careful thought into the design of their test, which is novel to the best of my knowledge.

Weaknesses: On p. 5, the authors claim that "a great representation cannot make up for the lack of a model and/or poor planning." I don't see how the results support such a broad claim, or even a narrow version of the claim. Where do the authors evaluate examples of a "great representation"? I'm actually not sure what that even means in this context. In Section 3, the authors focus on value iteration as their canonical model-based planning routine, but in practice this is intractable for most modern benchmarks. The MB-SU is a stochastic approximation, but is probably still much too inefficient on its own. In Section 5, the authors analyze a more advanced method (MuZero), but on the same toy problems. I just don't know how to generalize these results to more challenging problems. The model taxonomy was interesting, but it seems orthogonal to the evaluation issues that are the main focus of the paper.

Correctness: I don't see any issues with correctness.

Clarity: The paper is clearly written.

Relation to Prior Work: On p. 3, it is argued that a model-based agent "is able to make accurate predictions about future observations and reward for any hypothetical sequence of actions." This is referred to as the "neuroscience interpretation". But in the neuroscience literature since the Daw et al. (2011) paper, which appears to be the only neuroscience paper cited, there has been more attention paid to the issue of model misspecification. See for example Akam et al. (2016, Plos Computational Biology) and Feher da Silva & Hare (2020, Nature Human Behaviour). It also ignores the fact that humans are limited in their lookahead abilities, and this has led to hypotheses about various planning heuristics, including chunking (Huys et al., 2015, PNAS), model-free guidance (Cushman & Morris, 2015), and partial evaluation (Keramati et al., 2016, PNAS). Some of this literature is summarized in the book chapter by Kool, Cushman & Gershman (2018).

Reproducibility: Yes

Additional Feedback: p. 2: "model-base" -> "model-based" p. 3: "one of key behavioral characteristic" -> "one key behavioral characteristic" p. 3: "optimal policy task B" -> "optimal policy for task B" Figure 1 caption needs more explanation. It's hard to understand what the colors and arrows correspond to. p. 6: "while most of the interesting problems are sparse reward in nature." This is grammatically incorrect, but also an entirely subjective statement.


Review 3

Summary and Contributions: Standard model-based RL methods are evaluated on single-task sample efficiency. However, this metric is conflated with other challenges of RL in general, like representation learning. The authors propose a new evaluation metric to specifically assess/disentangle the model-based capacities of an algorithm, which they cal the Local Change Adaptation (LoCA) Regret. The LoCA Regret is defined on two tasks which only differ in a small part of state space, but have a completely different optimal policy. The experimentally show their ideas on a simulated grid world task, and on a MountainCar task, where MBRL algorithms are much faster to adapt to the second task. --- EDIT after author rebuttal --- I raised my vote after the author rebuttal and reviewer discussion. The top terminal frac tion cannot account for near optimal behaviour, and the principle may not be applicable in all kinds of tasks (it needs a well-defined end goal), but both topics are good directions for future work. The contribution of the paper as a better way to evaluate MBRL stands. As became clear from the reviewer discussion, the parameters of the MuZero experiment are not clear. To me, especially the d parameter remains vague, i.e., whether it is num_unroll_steps or num_simulations in the original code base (I guess the latter one is the one you really want to vary?) In any case, I would advise the authors to rewrite and clarify this section.

Strengths: - Evaluation of model-based RL methods is indeed an open question in literature, so I think the problem that the authors identify is very relevant. Current methods try to squeeze out all information on a single task, but this is not the type of setting in which MBRL will show its true benefit (it is indeed conflated with other challenges like representation learning). The problem identified in this paper is therefore very relevant, and the main idea of the solution (focus on transfer) is sound and well-motivated as well.

Weaknesses: – Sec 3: You method strongly depends on the ‘top-terminal fraction’. I see multiple potential problems: 1) what worries me most, is that it only measures optimality. What if my model-based agent adapts very fast to the new domain but reaches just below optimal performance. Then my MBRL method can be very effective, but the LoCA regret will still be very large. 2) You now define optimal as reaching a particular endpoint, but optimal also depends on how fast I do this (what if I first walk around every node in the grid before stepping into T2, is that optimal? Note that the regret at the bottom of P4 cannot correct for this, as it sums all timesteps and multiplies with the success fraction), 3) in more complicated tasks, it can be hard to determine the optimal behaviour, i.e., to even define the ‘top-terminal fraction’. – Sec 3: I am not sure whether you representation learning experiments really illustrate anything. You current inflate the space with extra noise dimensions. However, in the tabular setting, to overcome this issue you just need more data and additional training iterations. The point is that, if your pretraining phase 2 is long enough, you will always correct for the noise. It thereby does not really add anything. The problem is that your current set-up does not allow for any learning of a representation. A better set-up would give the noisy-augmented state to pretraining phase 1, let it learn a more compact representation (e.g., with a neural network), and then re-use these learned compact representations in pretraining phase 2 and evaluation. This way, your phase 2 and evalution do not have to deal with the representation challenge any more, this is already done in phase 1. But this does not work with a tabular model, since it cannot learn any representations, and cannot compress. Therefore, I don’t think the current experiments in Sec. 3 illustrate your idea. – Sec 4: This section seems to give a brief review of model-based reinforcement learning. While this is useful, I don’t think it really fits into the paper. You don’t have much space to fully discuss these issues, and there have been several reviews and framework of model-based RL, see references below. I also think your current discussion in Sec. 4 has some errors (e.g., L233: many Atari environments are stochastic, like Pacman) , and omits much work, see the mentioned reviews. You sort of cherry-picked six algorithms (I2A, AtreeC,World Models, PLANET, MuZero,PETS). Why these approaches? You could consider writing a MBRL survey, but then it probably requires a separate paper and deeper treatment. – The tasks you propose are very simple (a small gridworld, and MountainCar). I think it is not trivial to design new LoCA tasks in more complex problems, especially because the agent needs to be set to a restricted region where the dynamics have altered. Many environments will not permit this (like Atari, which has to be reset to the start), and it in any case requires a lot of handcoding. You shortly mention that your ideas naturally apply to more complicated tasks, but I do not see how yet. Smaller issues: – L37: “How do you define an ideal model based method?” What is ideal? Planning at every timestep until convergence? This is infeasible in larger problems right? – L93: I don’t see why Dyna (ref 16) is a good illustration of your statement. Dyna still learns a global value function (reactive), and plans (deliberative, although only single step). All MBRL agents with a value fuction and some type of planning have both types of decision-making. L95: ‘Given the neuroscience interpretation of a model-based system.’ This sentence deserves a reference. Are lines 96-101 your own definition? – I would mention that sample complexity is not the only type of standard performance measure. Computational complexity is another one. Model-based RL methods typically trade higher computational complexity for lower sample complexity. – L157: I don’t really understand the implicit normalization advantage. You mean that a fraction is between 0 and 1? You could easily rescale the reward to achieve the same effect on the return range, and leave out all discussion of the top-terminal fraction. It seems quite a hassle, while I don’t really see the benefit? – Figure 3: The right graph does not have blue or yellow lines. – Sec 3.2: How does your MB-VI implementation work. After every real environment step, we update the model and then do value iteration until convergence? That is computationally expensive planning right? – Sec 3.2: What type of exploration do you use? – L173: I don’t think you show the effect of poor representations or poor hyperparameters. Both the irrelevant features and the lower learning rate have a simple effect: you require more update iterations. But all these updates happen in pretraining phase 2. So if your pretraining phase 2 is long enough, then the model will still converge. That all full MB-VI methods have 0 LoCA regret shows that your pretraining phase 2 was really long, so the smaller learning rate and noisy features could converge as well. I don’t see a deeper message here. – Table 1: How can the model-free methods still have a relative gain? They do not use the model at all, do they? It can only stem from the fact that they have partially initialized value functions, but the relative gain will then strongly depend on your exploration hyperparameters. You do not mention anything about the exploration approach in the evaluate task B phase. – L187: I needed some time to think why the “relative gain” makes sense. I think you could motivate these definitions better. So gain describes for each method how much worse the method does without pretraining, and the relative gain allows us to compare between methods. – L240-251: Your motivation is a bit far stretched. The main aspect of MuZero is that is implicitly learns a model while optimizing for cumulative reward (non next-state prediction). Nearly all planning methods make multi-step predictions. – L240-251: I don’t understand how your 2-step and 5-step models look like? Are they conditioned on all the intermediate actions you take? Because if they are, I don’t understand why they would perform so badly, especially in a tabular model? In any case, like this the experiment does not give me any deeper insight, since it is not fully described, and you don’t explain your observations either. Why would a multi-step model hurt model-based performance? – Sec 5: I am not convinced the MountainCar set-up you propose is challenging. Swinging the underactuated car up is challenging, but your task B is to get it to the bottom, which is almost the attractor point of the system anyway. – Sec 5: Like in Sec. 3, I get the feeling you pick two hyperparameters to vary without much motivation. Why do you vary the hyperparameters of MuZero, but not the hyperparameters of SARSA? And what does the variation in depth and epsilon tell us about model-based transfer. Your conclusion of the section is only about the difference with SARSA right? – L299: “it is still a fair amount away from ideal model-based behaviour” – So what is the ideal model-based behaviour here? You should actually be able to assess that with you method right? Some MBRL surveys: Moerland, Thomas M., Joost Broekens, and Catholijn M. Jonker. "Model-based Reinforcement Learning: A Survey." arXiv preprint arXiv:2006.16712 (2020). Polydoros, Athanasios S., and Lazaros Nalpantidis. "Survey of model-based reinforcement learning: Applications on robotics." Journal of Intelligent & Robotic Systems 86.2 (2017): 153-173. Hester, Todd, and Peter Stone. "Learning and using models." Reinforcement learning. Springer, Berlin, Heidelberg, 2012. 111-141.

Correctness: I think the paper has some smaller gaps, as mentioned above in the Weaknesses section. In particular: - The definition of top-terminal fraction only looks at perfectly optimal behaviour, not near optimal behaviour. - I don’t think the representation learning challenge is properly set-up (it now only injects noise into the state at a tabular level, which does not permit for any representation learning in the pretraining phase 1) - The mini review of MBRL in Sec 4 is too shallow and does not really fit with the rest of the paper.

Clarity: Yes, the paper is clearly written.

Relation to Prior Work: The paper does not have a related work section, but I agree that this is not necessary. The authors do properly motivate in the introduction how standard model-based RL is evaluated, and why this is conflated with other RL challenges.

Reproducibility: Yes

Additional Feedback: Summary: This paper identifies a very relevant and interesting problem. Indeed, the way we usually evaluate model-based RL papers (sample complexity on a single task) is poor, since it is conflated with representation learning challenges, and has little to do with how humans use models to quickly adapt to slightly different tasks. I therefore really like the idea to focus on transfer between tasks as a metric to evaluate MBRL performance. However, I do not think the paper in its current form has too many gaps and loose ends (see above). I think the definitions of the top-terminal fraction has several problems (e.g., it only assess pure optimal behaviour, while the benefit of MBRL can also be close to optimal behaviour), I don’t think the representation learning challenge is properly set-up (it now only injects noise into the state at a tabular level, which does not permit for any representation learning in the pretraining phase 1), and the mini review of MBRL in Sec 4 is too shallow and does not really fit with the rest of the paper. Also, the set-up for the LoCA regret is quite complicated and not trivial to apply to more complex tasks. Once again, I do think this problem the authors identify is very relevant, but I believe the paper is right now in a too preliminary form.


Review 4

Summary and Contributions: This paper study a metric that describes how model based a method is based on a carefully designed two terminal state modified MDP.

Strengths: The author has shown very interesting results that highlights the benefit of a model, under the grid world environment. I.e. an oracle model based method could adapt very fast to slight changes of the environment even if some hyperparameters are wrongly set. The author has also shown interesting results that comparing muzero with model-free methods, and show that muzero has some level of model based benefits. They are all very nice and intuitive results and the authors have explained them very well.

Weaknesses: The only concern I have about this paper is that the author proposed a single metric that helps to debug model based methods. However, the authors haven't shown how to use this metric to further analyze and improve the model. The metric can reveal how "model based" an algorithm is, but as the author have mentioned, it is not clear whether we want to go all the way to model-based regeme since it seems like on-policy components, or model free components are necessary to achieve good performance. Given this, we might not want a model to have zero LoCA regret, since that method might needs a very complex planning compoment, resulting in a super long test running time. I would recommend the author to further study how to properly use this metric to analyze / improve a sample model, like muzero. Overall, I think the author has proposed a very interesting metric, however, it is still not super clear about how to use this metric. I recommend below acceptance at this time, but if the author can show some concrete example of the usage of this score, I'd love to change my rating.

Correctness: Yes, the method and claims are correct and is insightful. The empirical methodology is also correct.

Clarity: Yes, the paper is very well written

Relation to Prior Work: Yes, it has extensive discussion to previous works.

Reproducibility: Yes

Additional Feedback:

[Author Response · NeurIPS 2020]

We thank all reviewers for their time and appreciate the thoughtful feedback. And we are happy that all reviewers agree that the topic of our paper is both interesting and important. We only have space to address the main concerns below, but will take into account all feedback for the camera-ready version.

**R2: "I don't think this paper will be very impactful if it only shows results on a toy domain."** We respectfully disagree; in fact, we argue the opposite: the fact that we can show the limitations of MuZero on a simple domain like Mountain Car strenghtens our claims rather than weakens it. If MuZero is not able to get close to 0 LoCA regret on a trivial task like Mountain Car, it definitely won't be able to achieve this in more complex domains.

**R3: There are multiple issues with the top-terminal fraction. 1) It only measures optimality. 2) Optimal is defined as reaching an end-point, but optimal also depends on how fast this happens. 3) in complex tasks, it can be hard to define the top-terminal fraction.** Great points, but there is a small, crucial part in our definition of the top-terminal fraction to prevent precisely issues 1) and 2) mentioned here. We define the top-terminal fraction as the number of times the agent end up in terminal T2 *within a certain time limit*. We mention this on line 136, but admit it is somewhat hidden and will highlight this better in our next iteration of the paper. In our experiments, we have set the time limit at approximately 90% of the average time an optimal policy requires, starting from the evaluation initial-state distribution. Regarding point 3), as long as a meaningful variation with two terminal states can be constructed, a well-defined top-terminal fraction exists. Furthermore, see our relevant remarks at the end of this page.

**R4: "the authors haven't shown how to use this metric to further analyze and improve the model"** First, we'd like to push back on the implicit notion that identifying a problem is not a valuable contribution in and of itself; many influential papers do just that. Furthermore, note that we *do* perform analysis using the LoCA regret (w.r.t. planning hyperparameters), which leads us to the important observation that on-policy elements hurt the ability to quickly adapt. This provides guidance/clarity to the research community that new techniques should be investigated to get model-based methods that achieve both good performance in long-horizon tasks as well as fast adaptation.

**R1: "I can imagine model-based methods that adapt slowly and model-free methods that adapt fast."** You bring up a great point: sophisticated model-free methods can behave very similar to model-based methods. That's why the primary goal of the LoCA regret is not to try to identify the internal process a method uses, but to identify *useful behavior* that, according to neuroscience, is associated with model-based learning.

**R1: "The paper would be strengthened by a clear example where LoCA is able to distinguish between sample complexity improvement due to confounding factors vs effective planning".** Also, **R2: "Where do the authors evaluate examples of a great representation?"** and **R3: "I am not sure whether your representation learning experiments really illustrate anything.** These shared concerns have made us realize that the experiments from Section 3.3 should be better explained. We do believe these experiments are the right ones to show, but will add further explanation as to why these are relevant in the context of representation learning. In particular, we want to clarify the following: if method A uses a state-space with additional random features and method B uses a state-space without such features, then method B can be viewed as having a representation-learning module, compared to method A. Because if method B was given the same state-space as method A, but would also have a representation-learning module that learns to ignore the irrelevant random features during pretraining, the LoCA regret would be the same. So the comparison between, for example, regular Sarsa($\lambda$) (without random features) and MB-VI, $S_{mult} = 5$ (which has random features) can be viewed as two methods operating on the same state space, where one method uses no planning but has a representation-learning module, while the other has no representation-learning module, but uses planning. We hope this clarifies things.

**R3: "In the tabular setting, to overcome extra noise features in the state-space, you just need more data and training iterations. [...] If pretraining phase 2 is long enough, you always will correct for the noise."** Under the condition that a method can find a near-optimal policy in the limit (which holds for all our experiments), the difference between a poor and a good representation expresses itself *only* through data efficiency, also in non-tabular settings. Our LoCA pretraining is designed among others to remove the effect of the representation, as it is a confounding factor. Also, the effect is only removed for model-based methods; if a method is model-free (Sarsa($\lambda$)) or uses limited planning (MB-SU), the representation does effect the LoCA regret (see Table 1). This effect does *not* go away by having a long pretraining phase 2, because only a restricted part of the state-space is visited during this phase.

**R3: "I think it is not trivial to design new LoCA tasks in more complex problems, especially because the agent needs to set a restricted region [...]."** Even if designing new LoCA tasks would take some engineering effort, this does not substantially reduce the importance of this paper. Ultimately, there is no need to design a LoCA task for every possible domain; only a small set of representative domains is needed. Besides this, as long as a task-implementation gives a user the ability to set the state of the task, implementing a restricted region is straightforward: a wrapper around the task can be implemented that resets an agent to its previous state, as soon as an action moves the agent outside the restricted region, effectively giving such actions a 'no-op' effect.

[Meta-Review · NeurIPS 2020]

This paper proposes a method for identifying model-based behavior in RL agents (the “LoCA regret”), which can be used without knowing anything about the internal structure of the agent itself. This method is demonstrated to correctly distinguish between classical known model-free and model-based agents. It is also used to analyze MuZero, revealing that although MuZero is in principle a model-based algorithm, it does not make optimal use of its model. The reviewers agreed that the LoCA regret is a useful metric, and felt that doing careful evaluation of agents by designing metrics like this is an important area of research in RL. I agree, and found very interesting the demonstration that just because a particular algorithm makes use of a model, doesn't necessarily mean that the algorithm will have the properties that we think of as being associated with model-based algorithms. While there was some debate during the discussion period about some of the choices regarding the calculation of the LoCA regret (e.g. top-terminal fraction), the reviewers came to the agreement that the metric as presented is worthy of publication. Indeed, it was pointed out during the discussion that the fact that the paper generated so much discussion and follow-up questions is indicative of the interest it will draw if accepted. I therefore believe this work will be quite impactful and recommend acceptance. However, there was also a sense during the discussion that some of the experiments (specifically, those in Section 5) were unclear and potentially even somewhat misleading. For example, when asked to clarify what the variable ‘d’ corresponds to in the provided code, the authors replied that it corresponds to ‘num_unroll_steps’. While the paper states that ‘d’ controls the depth of MCTS, in the code ‘num_unroll_steps’ is actually a parameter governing how the model is trained. It sounds to me like the parameter the authors meant to vary in order to change the depth of search would be `num_simulations`. Similarly, the paper implies that Table 2 presents results with MuZero, but upon clarification, it seems like it is a different algorithm (though it is unclear what). Moreover, as R3 points out in their review, it is also not clear exactly what the 2-step and 5-step models look like or how they were trained. In general, a paper needs to be written so that all of these implementation details and design choices are clear and could be reproduced, and I do not feel Section 5 satisfies this criteria. I don’t believe that the issues with Section 5 detract so much from the paper as to warrant rejection, but as it stands, Section 5 is very unclear and as a result not particularly informative (aside from the main result that MuZero does not achieve zero LoCA regret). I therefore request that the authors rewrite Section 5 to be much clearer for the camera-ready version.